# A non-lethal stable isotope analysis of valued freshwater predatory fish using blood and fin tissues as alternatives to muscle tissue

Lukáš Vejřík[1], Ivana Vejříková[1]*, Zuzana Sajdlová[1‡], Luboš Kočvara[1‡],
Tomáš Kolařík[1‡], Daniel Bartoň[1‡], Tomáš Jůza[1‡], Petr Blabolil[1,2‡], Jiří Peterka[1‡],
Martin Čech[1‡], Mojmír Vašek[1]

1 Biology Centre of the Czech Academy of Sciences, Institute of Hydrobiology, České Budějovice, Czech
Republic, 2 Faculty of Science, University of South Bohemia in České Budějovice, České Budějovice, Czech
Republic

☯ These authors contributed equally to this work.
‡ ZS, LK, TK, DB, TJ, PB, JP, and MC also contributed equally to this work.
* ivana.vejrikova@hbu.cas.cz

doi.org/10.1371/journal.pone.0297070

UNITED STATES

**Data Availability Statement:** The dataset analyzed
during the study is provided in the Supporting
Materials.

## Abstract

Stable isotope analysis (SIA) is widely used to study trophic ecology and food webs in
aquatic ecosystems. In the case of fish, muscle tissue is generally preferred for SIA, and the
method is lethal in most cases. We tested whether blood and fin clips can be used as non-
lethal alternatives to muscle tissue for examining the isotopic composition of two freshwater
predatory fish, European catfish (*Silurus glanis*) and Northern pike (*Esox lucius*), species of
high value for many freshwater systems as well as invasive species in many others. Blood
samples from the caudal vein, anal fin clips, and dorsal muscle obtained by biopsy punch
were collected from four catfish and pike populations (14–18 individuals per population).
Subsequently, these samples were analyzed for $\delta^{13}$C and $\delta^{15}$N. The effects of alternative
tissues, study site, and fish body mass on the isotopic offset were investigated. Both species
showed a correlation between the isotopic offset and the tissue type, as well as the study
site, but no significant relationship with the body mass. The isotopic offsets between tissues
were used to calculate the conversion equations. The results demonstrated that both blood
and fin clips are suitable and less invasive alternative to muscle in SIA studies focused on
European catfish and Northern pike. Blood provided better correspondence to muscle iso-
tope values. However, our results clearly demonstrated that isotopic offsets between tissues
vary significantly among populations of the same species. Therefore, obtaining a muscle
biopsy from several individuals in any population is advisable to gain initial insights and
establish a possible population-specific inter-tissue conversion.

## Introduction

Stable isotopes are widely used in aquatic science [1–3], primarily $\delta^{13}$C and $\delta^{15}$N [3, 4]. They
are used in a broad spectrum of studies such as indicating nutrient pollution [5, 6],

**Funding:** 1) MČ: European Commission within the program of the LIFE21-NAT/IT/PREDATOR (project No. 101074458 – Life Predator). 2) PB: Applied Research Program of the Ministry of Agriculture – project "Methodology of predatory fish quantification in drinking-water reservoirs to optimize the management of aquatic ecosystems" (No. 461 QK1920011). 3) MČ and JP: Czech Academy of Sciences within the program of the Strategy AV 21 (project No. RP20 – Water for life, and RP21 – Land conservation and restoration). The funders had no role in study design, data collection and analysis, decision to publish, or preparation of the manuscript.

**Competing interests:** The authors have declared that no competing interests exist.

determining contaminant bioaccumulation [7, 8], tracking changes in carbon cycle over time [9], investigating aquatic food webs [10–12], and even assessing individual specialization [13]. Muscle tissue is commonly used for stable isotope analysis (SIA) of vertebrates [14], specifically the dorsal white muscle in fish studies [15]. Approximately 0.5 mg of dry mass is necessary for one analysis which constitutes a relatively substantial and often even lethal intrusion into the fish body [16]. Fish typically occupy higher positions within food webs underscoring the significance of their inclusion in the analysis of trophic structure and energy flow within aquatic ecosystems [17]. Nevertheless, the considerable size and high trophic position of predatory fish species impose constraints on sample sizes [12, 18]. Predatory fish exhibit notably lower population densities, and capturing large individuals through conventional sampling methods is usually challenging [12]. The substantial reduction of key species populations can also have a significant impact on ecosystem functioning [19]. Additionally, predatory fish are intentionally introduced into various systems (e.g., drinking water reservoirs) to exert a top-down effect on lower trophic levels. Therefore, it is preferable not to reduce their population density by lethal sampling [20]. In certain locations, such actions are even prohibited by the regulations of respective authorities.

The use of non-lethal sampling in stable isotope studies of fish, aimed at preventing unnecessary mortality of sampled animals, has recently received significant attention [16, 17, 21, 22]. Recent studies have explored non-lethal and minimally invasive alternatives, including scales, mucus and fins [17, 22–25]. The use of fin tissue has been the most prevalent [12], and such tissue is capable of regeneration [26, 27]. In contrast, collecting and employing blood for fish studies is relatively infrequent [28], despite its common application in various analyses, including SIA of higher vertebrates [29, 30]. The sampling is straightforward, particularly for large individuals, and results in minimal injury [13] compared to the muscle tissue sampling through biopsy punch [17, 31]. Although a biopsy itself is considered a non-lethal method for obtaining tissue, and according to Henderson et al. [31], it can even be performed in fish under 30 cm in size, the primary advantage of blood tissue collection over biopsy lies in the easier and gentler collection process. Additionally, there is a lower risk of introducing infections due to minimal external injury [32, 33]. This method is more comfortable for both the sampled individual and the person conducting the sampling. Despite the fact that biopsies often disrupt vessels in the muscle, causing bleeding in volumes greater than required for SIA blood analysis (personal observation), blood collection from fish results in complete healing within 2–3 weeks, demonstrating a 100% survival rate in teleost blood draws of 1 μL g$^{-1}$ with no post-treatment [34]. Approximately 200 μL of blood is sufficient for SIA, making this collection method safe for fish as small as 200 g of body mass [32, 34]. While it is possible to reduce the risk of introducing infections into biopsy wounds by using techniques such as applying Fish Bandage$^{TM}$ and a non-toxic, non-allergenic cellulose-based powder that forms a clear viscous gel upon contact with water and can treat skin ulcers and bind open wounds in fish when combined with an antiseptic [31], this represents an additional step in the sampling process, consequently increasing the sampling time.

Several studies have demonstrated correlations between isotopic signals from 'non-lethal tissues' and those from muscle tissue [16, 17, 22, 35, 36]. Nevertheless, it is essential to consider that the isotopic offsets, defined as the differences in isotope value between individual tissues, are not solely attributed to variations in tissue protein composition [37]. Stable isotope biokinetics also play a role in this regard [13, 38], and although often overlooked, they likely have a significant impact on differences among populations and various life stages within the same species [21, 39]. The biokinetics of stable isotopes in various tissue types, species, and environments is a complex issue. Unfortunately, current findings do not provide clear conclusions about the isotopic turnover rate of different tissues [13, 38, 40, 41]. Turnover depends not only

on tissue type but also on individual size and environmental temperature [38]. For an average organism weighing 1 mg, the isotopic half-life at 10°C is nine days, and at 40°C, it is only three days for both $^{13}$C and $^{15}$N (full turnover is about four to five times longer). However, for a 100 g organism, the half-life at 10°C is 80 days for $^{13}$C and 84 days for $^{15}$N. At 40°C, these values decrease to 28 days for $^{13}$C and 30 days for $^{15}$N. In the case of a 100 kg organism, the half-life at 10°C is 303 days for $^{13}$C and 321 days for $^{15}$N. At 40°C, these values decrease significantly to 105 days for $^{13}$C and 115 days for $^{15}$N [38]. As shown above, isotopic turnover rates vary among individual elements. It appears that $^{15}$N turnover rates are slightly longer than those of $^{13}$C [38, 40]. However, in the case of summer flounder (*Paralichthys dentatus*), no difference in the half-life between $^{15}$N and $^{13}$C was observed [41].

Based on several studies, the order of tissues in terms of isotopic half-life, from fastest to slowest, is approximately as follows (with slight variations in individual studies): Plasma → Liver → Fin → Heart → Mucus → Blood → Red blood cells → Bone collagen → Scale → Muscle [13, 38, 40, 41]. Another important factor that can influence isotopic turnover is the type of diet. It appears that different body tissues respond to dietary changes by altering their isotopic turnover rates. Tissues with longer isotopic turnover tend to exhibit greater differences depending on the type of diet [40].

Although the notable increase and growing trend towards non-invasive sampling are evident, there remains a lack of information concerning the relationship between isotope signatures of muscle and less harmful tissues for numerous species and populations [17, 22]. This study aims to contribute to the existing knowledge by examining the use of fin clips and blood in comparison to muscle tissue for two key apex predator fish species: European catfish (*Silurus glanis*) and Northern pike (*Esox lucius*) [12, 42]. The studied species are among the most widespread predatory fish in Europe and thus play a crucial role in maintaining ecosystem stability through top-down control [43]. To mitigate the influence of any single location, samples were collected from four distinct study sites. The primary aim was to investigate the feasibility of using fin and blood tissues as less harmful alternatives for $\delta^{15}$N and $\delta^{13}$C analyses in place of muscle tissue. The specific aims were as follows: (i) to examine the correlation between blood and fin tissues isotope signatures and those of muscle tissue, (ii) to assess the suitability of blood or fin tissue as a viable alternative, (iii) to develop appropriate conversion equations to enable the interchangeable use of different tissues in future stable isotope studies, and (iv) to explore whether variations in isotope signatures among tissues correlate with fish body mass differences.

## Methods

### Study design

Fishes were treated in accordance with the Experimental Animal Welfare Commission under the Ministry of Agriculture of the Czech Republic guidelines (Ref. No. CZ 01679). The work was approved by the Ethics Committee of the Czech Academy of Sciences.

The study was conducted in 2017 at four water bodies of similar size and fish species composition, but varying in their trophic states: two oligotrophic post-mining lakes, Most and Milada, and two meso-eutrophic reservoirs, Žlutice and Římov, Czech Republic (Fig 1). For basic parameters of the study sites see Table 1, and for further details see [20, 44].

Electrofishing (method description by [45]) and long-lines (method description by [20, 46, 47]) were used in 2017 to capture adult individuals of European catfish and Northern pike on 4–7 September in Most Lake, 11–14 September in Milada Lake, 25–28 July in Žlutice Reservoir, and 18–21 July in Římov Reservoir.

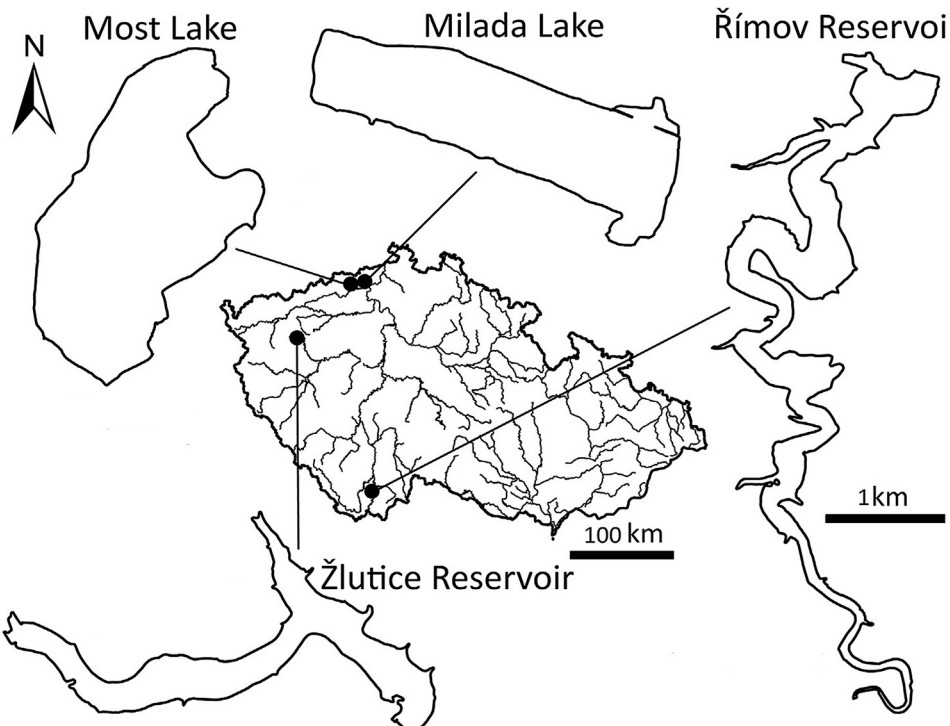

**Fig 1. Map of the Czech Republic with four study sites, Most and Milada artificial lakes, and Žlutice and Římov reservoirs.** Lakes and reservoirs share the same scale.

Body size and mass of European catfish ranged from 630 to 1,500 mm SL (mean±SD: 1046 ±214 mm) and 1.6 to 21.8 kg (mean±SD: 8.35±5.18 kg), and Northern pike body size ranged from 450 to 1,200 mm SL (729±202 mm), and mass ranged from 0.6 to 13.8 kg (3.75±3.06 kg). All individuals were weighed to the nearest gram, and samples of three body tissues were collected for SIA under anesthesia, a bath containing clove oil. Specifically, a small sample of the anal fin (1 cm$^2$, resection), 0.2 cm$^3$ of dorsal muscle (biopsy punch Miltex, skin removed), and 1 mL of blood from the caudal vein using an 18-gauge needle Sterican were taken. The three tissue samples from each individual were placed on ice and transferred to a laboratory freezer for SIA. Subsequently, all European catfish and Northern pike were released back into the water bodies.

## Stable isotope analysis (SIA)

All frozen samples for SIA were subsequently dried at 60°C for 48h and ground into a uniform powder using a Retsch MM 200 ball-mill (Retsch GmbH, Haan, Germany). Minor subsamples (0.52–0.77 mg) were carefully placed into tin cups for $\delta^{13}$C and $\delta^{15}$N analysis. All SIA were

**Table 1. Basic hydrological and geographical parameters of the four study sites.** Total phosphorus (TP) presents an average from longitudinal profiles of three areas sampled four times per year in 2013–2017.

| Site | Area (ha) | Volume ($10^6$ m$^3$) | Long-term inflow (m$^3$ s$^{-1}$) | Retention time (days) | Max. depth (m) | Altitude (m a.s.l.) | Trophy, TP (µg L$^{-1}$) |
|---|---|---|---|---|---|---|---|
| Most | 311 | 70 | 0.06 | drainless | 75 | 199 | 2.5 |
| Milada | 250 | 36 | 0.04 | 10,248 | 25 | 145 | 5 |
| Žlutice | 161 | 16 | 1.24 | 146 | 23 | 508 | 25.8 |
| Římov | 210 | 34 | 4.38 | 93 | 45 | 468 | 26.4 |

performed using a FlashEA 1112 elemental analyzer coupled to a Thermo Finnigan DELTA-plus Advantage mass spectrometer (Thermo Fisher Scientific Corporation, USA) at the University of Jyväskylä, Finland. Carbon and nitrogen isotope ratios are presented as $\delta^{13}C$ and $\delta^{15}N$ values relative to the international standards for carbon (Vienna PeeDeeBelemnite, Austria) and nitrogen (atmospheric nitrogen). The precision of the analytical results was determined to be ±0.20‰ through repeated analyses of a consistent standard (Northern pike white muscle tissue), which was inserted in each analytical run following every five samples. The data were not adjusted for lipid content due to elemental carbon–nitrogen (C:N) ratios observed in all tissues being less than 4 (except for a single catfish blood sample at Římov Reservoir) indicating a low lipid content [48].

## Data analysis

Prior the analyses, the data distribution was visually checked by histogram plotting. No obvious skewness distribution was observed. Paired t-tests were conducted to assess whether blood and fin isotope values differ from those of muscle, and also to determine whether C:N ratios in blood and fin of individuals at each study site differ compared to those muscle tissue. The isotopic offsets between tissues, defined as the differences between blood and muscle or fin and muscle stable isotope values, were compared among study sites (Milada, Most, Žlutice, Římov) for European catfish and Northern pike using a one-way repeated measures analysis of variance (ANOVA) with multiple comparisons (Tukey HSD test). Additionally, the one-way ANOVA was used to assess the differences in C:N ratios among all tissue types across all study sites. The statistical analyses were performed in STATISTICA 9.1 [49] and used the standard level of significance of α = 0.05. Linear model with the random effect of site was applied to model the relationships between mussel isotopes with the isotopes in fin and blood and between the isotope offsets and fish mass. The models were prepared for each species and C and N isotope separately. After fitting the models, regression diagnostics was applied by plotting the residual vs. fitted values and Normal QQ Plot to check the residuals variance. The models were developed using R packages "lme4" [50] and "nlme" [51] in R software [52].

## Results

The $\delta^{13}C$ and $\delta^{15}N$ values in blood and fins of European catfish and Northern pike were generally found to be significantly different from those in their muscle tissue. However, the difference in $\delta^{13}C$ between the blood and muscle was not significant for European catfish at one study site and for Northern pike at two study sites. Regarding $\delta^{15}N$, there was no significant difference between the blood and muscle of Northern pike in two study sites, and between the fin and muscle of Northern pike in one study site (Table 2).

Blood generally showed neither clear enrichment nor depletion of δ13C compared to muscle tissue. In European catfish, the blood exhibited a slight $\delta^{13}C$ enrichment at three study sites and a slight depletion at one study site. For Northern pike, the blood displayed slight δ13C enrichment at only one study site and slight depletion at three study sites. The fin was consistently and significantly enriched in $\delta^{13}C$ in comparison to the muscle tissue in both species across all study sites (Table 2). In both European catfish and Northern pike, both the blood and fins were consistently and significantly enriched in δ15N when compared to the muscle tissue, observed across most study sites. The degree of these differences was more pronounced in the blood compared with fins of European catfish. In the case of Northern pike, on average, the degree of these differences between blood and fin was similar. The difference in both $\delta^{13}C$ and $\delta^{15}N$ between less harmful tissues and muscle tissue tended to be greater in European catfish when compared to Northern pike (Table 2).

**Table 2. Mean ± SE values of δ¹³C and δ¹⁵N, in muscle (M), blood (B) and fin (F) tissues of European catfish (top) and Northern pike (bottom) at four study sites (Most, Milada, Žlutice and Římov).** Significant differences between offset (means how given tissue is enriched or depleted compared to muscle tissue) of blood and muscle (B) and fin and muscle (F) (paired t-test) were examined. The C:N ratios were measured in three tissues of European catfish (top) and Northern pike (bottom). Asterisks indicate a significance (paired t-test, $p > 0.05$ = -, $p < 0.05$ = *, $p < 0.01$ = **, $p < 0.001$ = ***).

| Site | Tissue | Mean ± SE δ¹³C (‰) | δ¹³C offset | Statistics t | p | Mean ± SE δ¹⁵N (‰) | δ¹⁵N offset | Statistics t | p | C:N |
|---|---|---|---|---|---|---|---|---|---|---|
| Most | M | -22.86 ±0.96 | | | | 15.95 ± 1.01 | | | | 3.28 ± 0.05 |
| (N = 16) | B | -22.36 ±1.27 | 0.5 ± 0.53 | 3.72 | ** | 16.35 ± 1.08 | 0.40 ± 0.48 | 3.25 | ** | 3.62 ± 0.12 |
| | F | -21.46 ± 1.02 | 1.4 ± 0.46 | 11.88 | *** | 16.31 ± 1.03 | 0.36 ± 0.46 | 3.05 | ** | 3.47 ± 0.08 |
| Milada | M | -22.45 ± 0.77 | | | | 27.39 ± 0.66 | | | | 3.22 ± 0.09 |
| (N = 16) | B | -22.03 ±0.70 | 0.42 ± 0.45 | 3.64 | ** | 28.57 ± 0.53 | 1.18 ± 0.33 | 13.71 | *** | 3.62 ± 0.15 |
| | F | -21.38 ± 0.69 | 1.07 ± 0.45 | 9.22 | *** | 28.17 ± 0.53 | 0.79 ± 0.38 | 8.09 | *** | 3.40 ± 0.08 |
| Žlutice | M | -26.32 ± 0.67 | | | | 16.00 ± 1.41 | | | | 3.59 ± 0.34 |
| (N = 18) | B | -26.39 ± 0.66 | -0.07 ± 0.45 | 0.68 | - | 16.69 ± 1.23 | 0.70 ± 0.42 | 6.86 | *** | 3.63 ± 0.09 |
| | F | -25.71 ± 0.71 | 0.61 ± 0.49 | 5.15 | *** | 16.83 ± 1.20 | 0.83 ± 0.65 | 5.29 | *** | 3.52 ± 0.10 |
| Římov | M | -23.69 ± 1.75 | | | | 14.65 ± 0.47 | | | | 3.53 ± 0.31 |
| (N = 16) | B | -22.44 ± 0.37 | 1.26 ± 1.47 | 3.31 | ** | 15.13 ± 0.24 | 0.48 ± 0.29 | 6.59 | *** | 3.69 ± 0.41 |
| | F | -22.05 ± 0.31 | 1.64 ± 1.56 | 4.08 | *** | 15.05 ± 0.30 | 0.40 ± 0.31 | 4.98 | *** | 3.53 ± 0.14 |
| Most | M | -20.97 ± 1.43 | | | | 16.82 ± 0.78 | | | | 3.28 ± 0.02 |
| (N = 14) | B | -20.99 ± 1.26 | -0.02 ± 0.32 | 0.27 | - | 17.25 ± 0.86 | 0.43 ± 0.29 | 5.42 | *** | 3.63 ± 0.15 |
| | F | -20.22 ± 1.30 | 0.75 ± 0.66 | 4.09 | ** | 17.15 ± 0.74 | 0.34 ± 0.26 | 4.61 | *** | 3.50 ± 0.06 |
| Milada | M | -22.17 ± 0.89 | | | | 28.90 ± 0.72 | | | | 3.27 ± 0.03 |
| (N = 16) | B | -22.00 ± 1.05 | 0.18 ± 0.29 | 2.36 | ** | 29.05 ± 0.65 | 0.15 ± 0.31 | 1.87 | - | 3.57 ± 0.08 |
| | F | -21.07 ± 1.04 | 1.10 ± 0.30 | 14.30 | *** | 29.37 ± 0.70 | 0.47 ± 0.48 | 3.76 | *** | 3.49 ± 0.04 |
| Žlutice | M | -27.24 ± 0.64 | | | | 15.44 ± 0.95 | | | | 3.29 ± 0.05 |
| (N = 14) | B | -27.32 ± 0.63 | -0.08 ± 0.22 | 1.39 | - | 15.59 ± 0.82 | 0.15 ± 0.33 | 1.66 | - | 3.66 ± 0.18 |
| | F | -26.74 ± 0.50 | 0.50 ± 0.31 | 5.90 | *** | 15.44 ± 0.96 | 0.00 ± 0.13 | 0.02 | - | 3.55 ± 0.06 |
| Římov | M | -21.96 ± 0.33 | | | | 14.90 ± 0.31 | | | | 3.34 ± 0.07 |
| (N = 16) | B | -22.17 ± 0.41 | -0.21 ± 0.21 | 3.93 | ** | 15.00 ± 0.37 | 0.10 ± 0.17 | 2.27 | * | 3.61 ± 0.12 |
| | F | -21.63 ± 0.48 | 0.33 ± 0.28 | 4.70 | *** | 15.44 ± 0.49 | 0.54 ± 0.30 | 7.04 | *** | 3.56 ± 0.04 |

For European catfish, both δ¹³C and δ¹⁵N offsets between both blood and muscle, and fin and muscle significantly differed among study sites. The statistical values for δ¹³C offset were: ANOVA: $F_{3,62} = 6.87$, $p < 0.001$ for blood vs. muscle, and ANOVA: $F_{3,62} = 4.26$, $p < 0.01$ for fin vs. muscle. The statistical values for δ¹⁵N offset were: ANOVA: $F_{3,62} = 12.31$, $p < 0.001$ for blood vs. muscle, and ANOVA: $F_{3,62} = 4.30$, $p < 0.01$ for fin vs. muscle.

Also for Northern pike, the differences in both δ¹³C and δ¹⁵N offsets between both blood and muscle, and fin and muscle were significant among study sites. The statistical values for δ¹³C offset were: ANOVA: $F_{3,56} = 5.67$, $p < 0.01$ for blood vs. muscle, and ANOVA: $F_{3,56} = 9.77$, $p < 0.001$ for fin vs. muscle. The statistical values for δ¹⁵N offset were: ANOVA: $F_{3,56} = 3.81$, $p < 0.05$ for blood vs. muscle, and ANOVA: $F_{3,56} = 7.44$, $p < 0.001$ for fin vs. muscle.

Tukey HSD comparisons of pairwise offset differences between study sites revealed significant distinction. Specifically, there were significant differences in δ¹³C of European catfish between blood and muscle in two cases, between fin and muscle in one case. Additionally, for δ¹⁵N, significant differences were between blood and muscle in three cases and between fin and muscle in one case (Table 3). For Northern pike, Tukey HSD comparisons of pairwise offset differences between study sites revealed significant differences in δ¹³C between blood and muscle in one case, between fin and muscle in three cases, and for δ¹⁵N between blood and muscle in one case and between fin and muscle in two cases (Table 3).

**Table 3. Tukey HSD statistical values (p) of the difference in $\delta^{13}C$ and $\delta^{15}N$ offsets between blood and muscle (B—M), and fin and muscle (F—M) among study sites (Most, Milada, Žlutice, Římov).**

| Species | Offset: | $\delta^{13}C$ (B—M) | $\delta^{13}C$ (F—M) | $\delta^{15}N$ (B—M) | $\delta^{15}N$ (F—M) |
|---|---|---|---|---|---|
| European Catfish | Most × Milada | 0.992 | 0.727 | **<0.001** | 0.075 |
| | Most × Žlutice | 0.217 | 0.058 | 0.153 | **<0.05** |
| | Most × Římov | 0.072 | 0.871 | 0.937 | 0.994 |
| | Milada × Žlutice | 0.350 | 0.440 | **<0.01** | 0.992 |
| | Milada × Římov | **<0.05** | 0.283 | **<0.001** | 0.131 |
| | Římov × Žlutice | **<0.001** | **<0.01** | 0.423 | 0.060 |
| Northern Pike | Most × Milada | 0.197 | 0.120 | 0.052 | 0.711 |
| | Most × Žlutice | 0.937 | 0.419 | 0.066 | 0.053 |
| | Most × Římov | 0.237 | **<0.05** | **<0.05** | 0.336 |
| | Milada × Žlutice | 0.055 | **<0.01** | 0.999 | **<0.01** |
| | Milada × Římov | **0.001** | **<0.001** | 0.965 | 0.915 |
| | Římov × Žlutice | 0.563 | 0.692 | 0.965 | **<0.001** |

The C:N ratios were, on average, lowest in the muscle tissue of both European catfish and Northern pike. In contrast, blood reached the highest average values (Fig 2). In the case of European catfish, both blood and fin reached significantly higher C:N values than muscle in Milada and Most lakes (paired t-test, p < 0.001). Also in Žlutice Reservoir, C:N ratio was significantly higher in blood than in muscle of European catfish (paired t-test, p < 0.05), however no difference was found between fin and muscle (paired t-test $t_{17}$ = -0.75, p = 0.46). In Římov Reservoir, there was no difference in C:N ratio between both blood and muscle (paired t-test $t_{15}$ = -0.80, P = 0.44) and fin and muscle (paired t-test $t_{15}$ = -0.04, p = 0.97) of European catfish. For Northern pike, the C:N ratio was consistently and significantly higher in both blood and fin compared to muscle, across all study sites (paired t-test, p < 0.001) (Fig 2). Among study sites, C:N ratios were significantly different in muscle and fin of European catfish (muscle: ANOVA: $F_{3,62}$ = 7.98, p < 0.001; fin: ANOVA: $F_{3,62}$ = 5.69, p < 0.001) and Northern pike (muscle: ANOVA: $F_{3,56}$ = 6.30, p < 0.001; fin: ANOVA: $F_{3,56}$ = 6.60, p < 0.001). In contrast, there was no significant difference in C:N ratio in blood among the study sites either for European catfish (ANOVA: $F_{3,62}$ = 0.16, p = 0.92) or Northern pike (ANOVA: $F_{3,56}$ = 1.03, p = 0.39).

The linear regression revealed significant relationships between the $\delta^{15}N$ and $\delta^{13}C$ values in muscle, fin and blood of both European catfish and Northern pike (Figs 3 and 4). Thus, both less harmful tissues served as reliable predictors for muscle. Specifically, blood demonstrated strong predictive capabilities for both $\delta^{13}C$ and $\delta^{15}N$ values in the muscle of both European catfish and Northern pike. For European catfish, the coefficients of determination ($R^2$) for $\delta^{13}C$ were, on average, 0.83 and 0.79 for blood and fin, respectively (Fig 3). Similarly, for $\delta^{15}N$, the $R^2$ values were 0.91 and 0.84 for blood and fin, respectively (Fig 4). Regarding Northern pike, the $R^2$ values for $\delta^{13}C$ averaged 0.94 and 0.86 for blood and fin, respectively (Fig 3), and for $\delta^{15}N$ the $R^2$ values were 0.92 and 0.88 in blood and fin, respectively (Fig 4). The regression diagnostics of isotope linear model with the random effect of site demonstrate uniform residuals variance and lack of high leverage effects. Models intercepts and standard errors in fixed effects were higher for European catfish compared to Northern pike and slopes were comparable with exception of $\delta^{13}C$ for European catfish with lower slope (Table 4). Similarly, random effects were higher for European catfish compared to Northern pike (Table 4).

No distinct trends were found through linear regression analysis between isotopic tissue offsets and fish body mass in either species or in any study site. For European catfish, the $R^2$

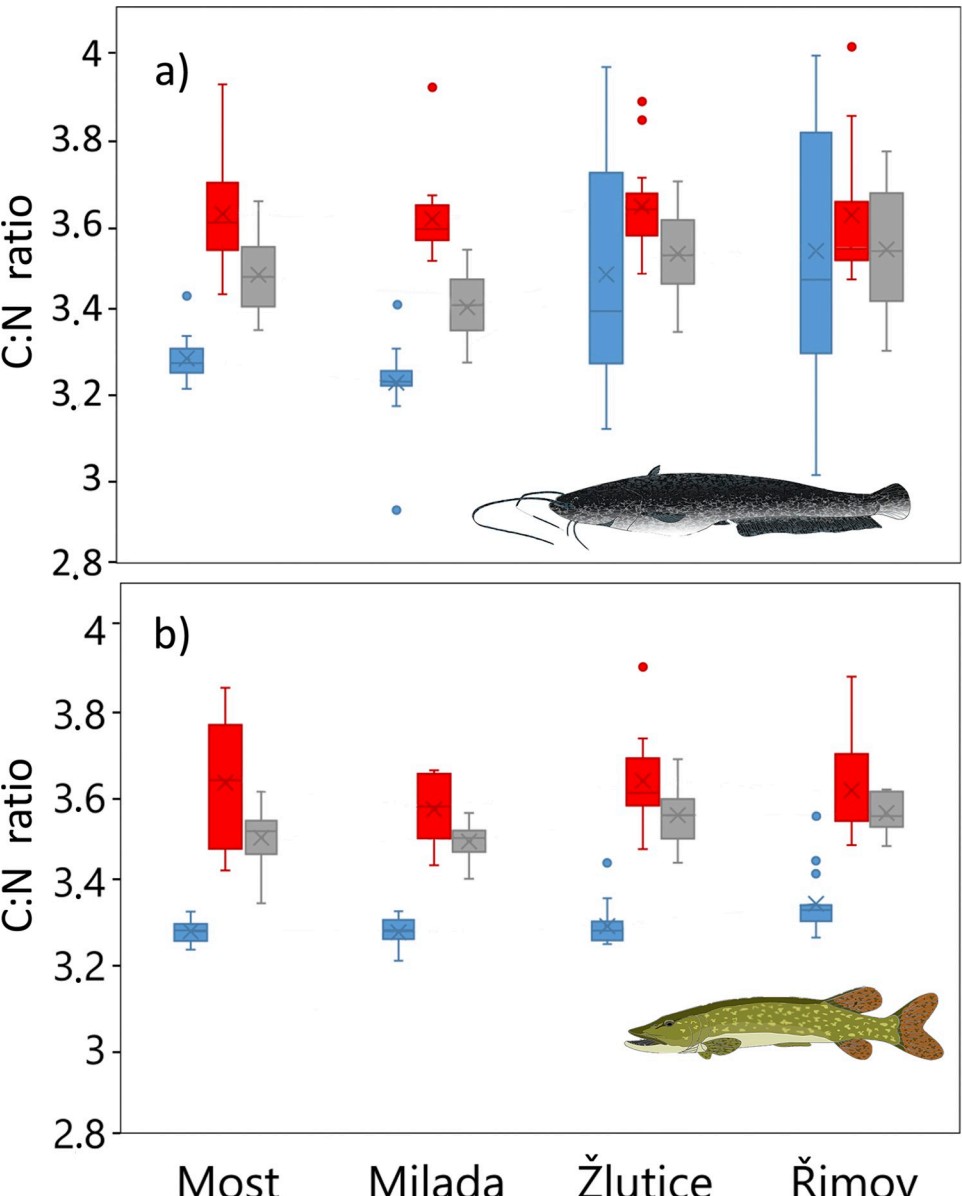

**Fig 2.** C:N ratios of European catfish (a) and Northern pike (b) in tissues (muscle: blue, fin: gray, blood: red) at four study sites (Most, Milada, Žlutice, Římov). Box and whiskers plots: upper and lower quartiles (boxes), median values (line inside the boxes), mean value (crosses), maximum and minimum values (whiskers), and outliers (circles) are shown.

reached mean values of 0.11 ± 0.17 SD for the blood and muscle $\delta^{13}C$ offset, and 0.17 ± 0.17 SD for the fin and muscle $\delta^{13}C$ offset in relation to the body mass. Regarding $\delta^{15}N$, the $R^2$ reached mean values of 0.24 ± 0.22 SD for the blood and muscle offset, and 0.05 ± 0.04 SD for the fin and muscle offset in relation to the body mass. For Northern pike, the $R^2$ reached mean values of 0.10 ± 0.11 SD for the blood and muscle $\delta^{13}C$ offset, and 0.01 ± 0.02 SD for the fin and muscle $\delta^{13}C$ offset. In the case of $\delta^{15}N$, the $R^2$ reached mean values of 0.07 ± 0.07 SD for the blood and muscle offset, and 0.04 ± 0.05 SD for the fin and muscle offset in relation to the body mass. The regression diagnostics of offset isotope linear model with the random effect of site demonstrate uniform residuals variance and lack of high leverage effects. Models'

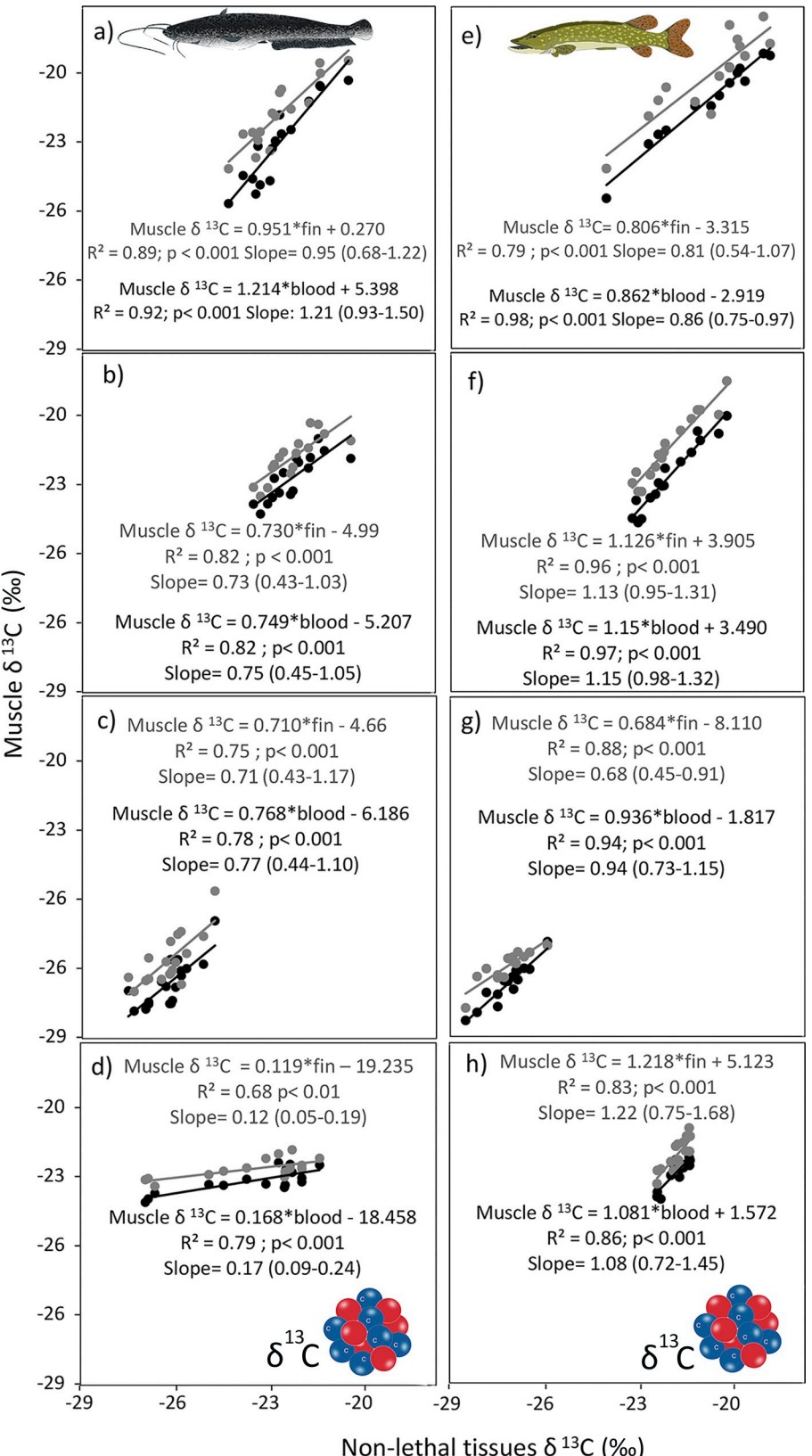

**Fig 3.** Linear regressions between $\delta^{13}$C of less harmful tissues (blood: black circles and line, fin: gray circles and line) and muscle tissue for European catfish (left column) and Northern pike (right column) in Most lake (a, e), Milada lake (b, f), Žlutice reservoir (c, g) and Římov reservoir (d, h). The same scale is maintained on both axes in all images. The linear regression equations for European catfish and Northern pike to convert $\delta^{13}$C values in blood (black equations) and fin (gray equations) to those in muscle are presented with added statistical significances of the linear regression and the 95% confidence intervals of the slopes.

intercepts were the highest for European catfish offset $\delta^{15}$N for all tissues combination, offset $\delta^{13}$C fin to body mass and Northern pike offset $\delta^{13}$C fin to body mass comparison (Table 5). The slopes values were generally low with the highest values for European catfish offset $\delta^{13}$C. The random effect intercept and residual variances were the highest for European catfish offset $\delta^{13}$C (Table 5).

## Discussion

Sampling white muscle typically involves sacrificing the fish [16]. However, in our case, none of the predatory fish were directly sacrificed. We employed a biopsy punch to extract the muscle, a method feasible for larger fish species [33]. Nevertheless, this procedure still constitutes a relatively invasive intervention into the fish body, and the potential consequences upon its release into the water are generally unknown [31]. In contrast, methods such as fin clipping [22, 53] and blood collection [54] are significantly less disruptive. Blood sampling through the caudal vasculature is widely employed in fish biology for investigating health and physiology [54]. In live fish, it offers a rapid, uncomplicated, and comparatively less harmful approach for obtaining tissue, in contrast to the invasive and even more intricate nature of biopsies [31, 54]. The fin clip collection is a routine and widely used technique, employed either to acquire tissue samples for various analyses [35] or to mark fish [55].

In general, the differences in $\delta^{13}$C and $\delta^{15}$N isotope values between muscle and less harmful tissues in both predatory fish were smaller in the case of blood than in the fin. This is likely to be due to a more comparable protein composition between muscle and blood than between muscle and fin [37]. However, it appears that this effect is not related to the biokinetics of stable isotopes.

In our previous study [13], we calculated isotopic half-life [38] for European catfish and Northern pike in the presented study sites. We considered both the mean mass of the individuals and the temperature at which they were found, as determined by telemetry measurements [42]. Based on these calculations, the average isotopic half-life for blood and fin were quite similar in both species. Conversely, the half-life of these tissues in both species significantly differed from the notably longer half-life of muscle. For European catfish, the average isotopic half-life values were 39, 44 and 153 days for blood, fin, and muscle, respectively. In the case of Northern pike, the values were 39, 41 and 139 days for blood, fin and muscle [13]. However, as mentioned in the introduction, the issue of biokinetics is highly complex and can be influenced by various other factors, such as the diet composition, which is unknown in this case [40]. Nevertheless, significant variations in both δ13C and δ15N were observed in most instances between muscle and blood, or between muscle and fin. The fin was enriched in $\delta^{13}$C compared to muscle in both European catfish and Northern pike across all study sites. This pattern is consistent with findings from several other studies focusing on diverse fish species [22, 53, 56] as well as various Northern pike populations [57, 58]. The fin tissue was enriched also in $\delta^{15}$N in both species at nearly all study sites. This is noteworthy since many previous studies have reported $\delta^{15}$N depletion in fin tissue compared to muscle in other species [22, 53] or have identified an inconclusive trend between fin and muscle [17, 33, 56]. However, for example, in the case of callop (*Macquaria ambigua*) in the Australian Darling River, the fins

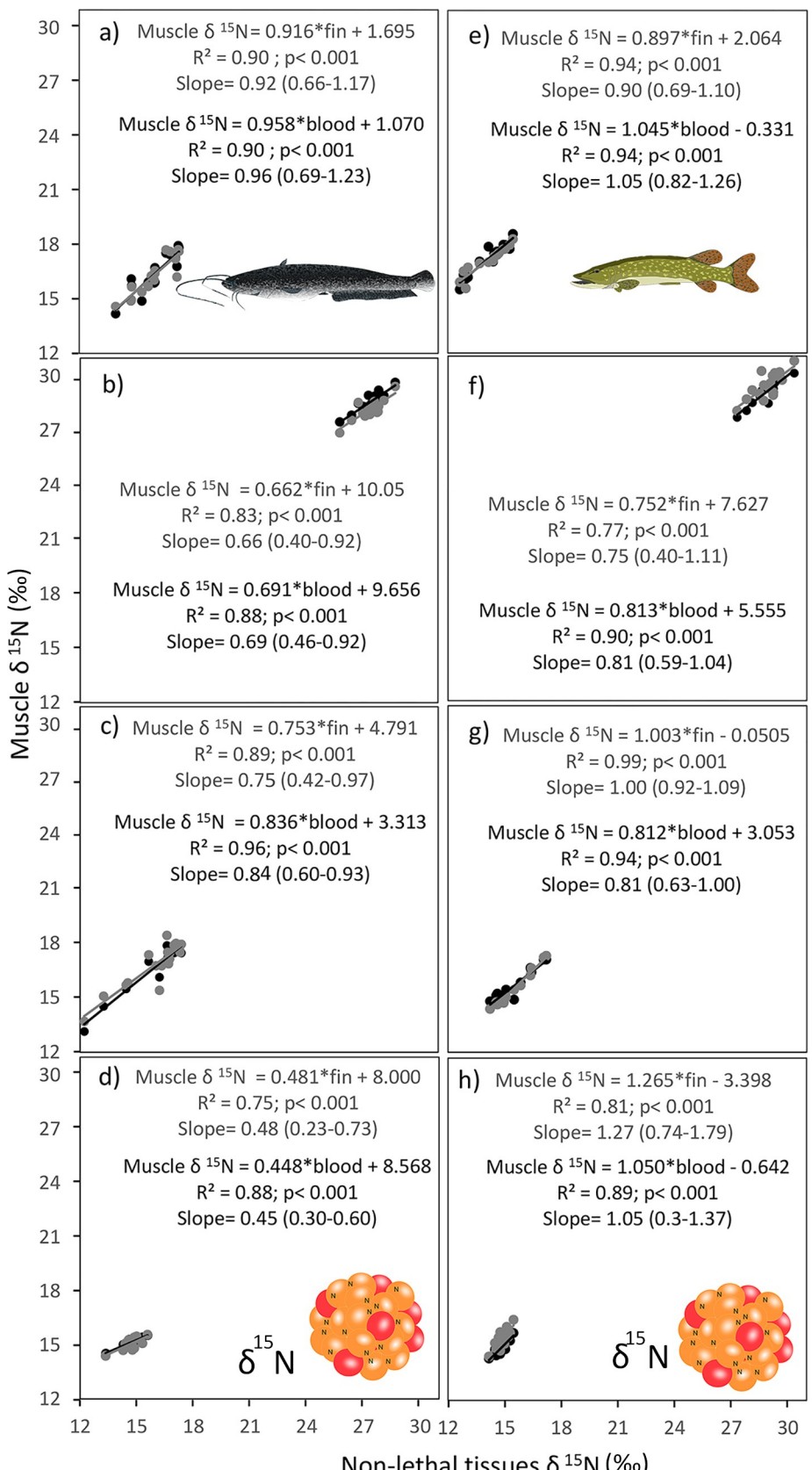

a) Muscle δ $^{15}$N = 0.916*fin + 1.695
$R^2$ = 0.90 ; p< 0.001
Slope= 0.92 (0.66-1.17)

Muscle δ $^{15}$N = 0.958*blood + 1.070
$R^2$ = 0.90 ; p< 0.001
Slope= 0.96 (0.69-1.23)

e) Muscle δ $^{15}$N = 0.897*fin + 2.064
$R^2$ = 0.94; p< 0.001
Slope= 0.90 (0.69-1.10)

Muscle δ $^{15}$N = 1.045*blood - 0.331
$R^2$ = 0.94; p< 0.001
Slope= 1.05 (0.82-1.26)

b) Muscle δ $^{15}$N = 0.662*fin + 10.05
$R^2$ = 0.83; p< 0.001
Slope= 0.66 (0.40-0.92)

Muscle δ $^{15}$N = 0.691*blood + 9.656
$R^2$ = 0.88; p< 0.001
Slope= 0.69 (0.46-0.92)

f) Muscle δ $^{15}$N = 0.752*fin + 7.627
$R^2$ = 0.77; p< 0.001
Slope= 0.75 (0.40-1.11)

Muscle δ $^{15}$N = 0.813*blood + 5.555
$R^2$ = 0.90; p< 0.001
Slope= 0.81 (0.59-1.04)

c) Muscle δ $^{15}$N = 0.753*fin + 4.791
$R^2$ = 0.89; p< 0.001
Slope= 0.75 (0.42-0.97)

Muscle δ $^{15}$N = 0.836*blood + 3.313
$R^2$ = 0.96; p< 0.001
Slope= 0.84 (0.60-0.93)

g) Muscle δ $^{15}$N = 1.003*fin - 0.0505
$R^2$ = 0.99; p< 0.001
Slope= 1.00 (0.92-1.09)

Muscle δ $^{15}$N = 0.812*blood + 3.053
$R^2$ = 0.94; p< 0.001
Slope= 0.81 (0.63-1.00)

d) Muscle δ $^{15}$N = 0.481*fin + 8.000
$R^2$ = 0.75; p< 0.001
Slope= 0.48 (0.23-0.73)

Muscle δ $^{15}$N = 0.448*blood + 8.568
$R^2$ = 0.88; p< 0.001
Slope= 0.45 (0.30-0.60)

h) Muscle δ $^{15}$N = 1.265*fin - 3.398
$R^2$ = 0.81; p< 0.001
Slope= 1.27 (0.74-1.79)

Muscle δ $^{15}$N = 1.050*blood - 0.642
$R^2$ = 0.89; p< 0.001
Slope= 1.05 (0.3-1.37)

Muscle δ $^{15}$N (‰)

Non-lethal tissues δ $^{15}$N (‰)

$δ^{15}$N

$δ^{15}$N

**Fig 4.** Linear regressions between $\delta^{15}N$ of less harmful tissues (blood: black circles and line, fin: gray circles and line) and muscle tissue for European catfish (left column) and Northern pike (right column) in Most lake (a, e), Milada lake (b, f), Žlutice reservoir (c, g) and Římov reservoir (d, h). The same scale is maintained on both axes in all images. The linear regression equations for European catfish and Northern pike to convert $\delta^{13}C$ values in blood (black equations) and fin (gray equations) to those in muscle are presented with added statistical significances of the linear regression and the 95% confidence intervals of the slopes.

were also enriched in $\delta^{15}N$ compared to the muscle [17]. A similar pattern was observed with the bleak (*Alburnus alburnus*) in Spain [53]. Therefore, it appears that, unlike $\delta^{13}C$, the trend in $\delta^{15}N$ is not as clear and may vary among species. Notably, Syväranta et al. [59] determined that the $\delta^{15}N$ of European catfish fin tissue exhibited a slight but insignificant depletion in relation to muscle tissue. However, in order to compare isotopic signatures between tissues, these authors aggregated data from catfish individuals collected across four distinct locations. Consequently, it became impractical to draw conclusions about population-specific disparities in $\delta^{13}C$ and $\delta^{15}N$ between muscle and fin tissues. The two prior studies on Northern pike that investigated the isotopic relationship between fin and muscle [58, 60] revealed that fin $\delta^{15}N$ was significantly depleted compared to muscle, whereas our observations demonstrated an opposing fin-muscle $\delta^{15}N$ difference in three out of four pike populations (with no significant difference between the two tissues in $\delta^{15}N$ within one studied population). Our findings undeniably illustrate that isotopic offsets between tissues can considerably differ among populations of the same species. Thus, it is unrealistic to assume the feasibility of devising a singular species-specific conversion equation that would universally apply to all populations of the given species. Rather, aligning with the perspectives of other researchers [39, 61, 62], we propose that a tissue-to-tissue conversion equation should be developed distinctly for each specific population of interest. The isotopic signal of individuals and their various tissues partly reflects the distinct protein composition of the tissues [37]. However, it is primarily influenced by the dietary composition of individuals or the entire populations. This diet composition can vary considerably within a species, both seasonally and across different localities [11–13]. The diet of apex predators in our study sites was extensively examined, based on both SIA and the stomach content analysis [12, 63]. The study sites differed significantly from each other in terms of the diet composition of apex predators and their growth rates [12, 63]. In the oligotrophic sites of Milada and Most, apex predators exhibited significantly slower growth compared to the meso-eutrophic reservoirs Žlutice and Římov [63]. Moreover, at Milada and Most lakes, the diet of apex predators displayed high variety, including invertebrates, fish, and semi-aquatic prey such as waterfowl, mammals, and amphibians. In contrast, at the Římov and Žlutice,

**Table 4. Parameters and coefficients of isotope linear model with the random effect of site.**

| Model parameters | | | Fixed effect | | | | Random effect | | | |
|---|---|---|---|---|---|---|---|---|---|---|
| Species | Isotope | Comparison | Intercept value | Intercept std. error | Slope value | Slope std. error | Intercept variance | Intercept std. dev. | Residual variance | Residual std. dev. |
| European catfish | $\delta^{13}C$ | Muscle -blood | -3.7903 | 2.1450 | 0.8598 | 0.0914 | 0.1652 | 0.4065 | 0.7404 | 0.8605 |
| | | Muscle-fin | -4.2773 | 1.8306 | 0.8634 | 0.0804 | 0.0692 | 0.2630 | 0.7966 | 0.8925 |
| | $\delta^{15}N$ | Muscle-blood | 0.2875 | 0.2653 | 0.9489 | 0.0133 | 0.0119 | 0.1091 | 0.1672 | 0.4089 |
| | | Muscle-fin | -0.1803 | 0.4627 | 0.9779 | 0.0233 | 0.0522 | 0.2285 | 0.2484 | 0.4984 |
| Northern pike | $\delta^{13}C$ | Muscle-blood | -0.5306 | 0.6249 | 0.9755 | 0.0268 | 0.0278 | 0.1668 | 0.0736 | 0.2712 |
| | | Muscle-fin | -2.3728 | 1.0239 | 0.9242 | 0.0451 | 0.1068 | 0.3268 | 0.1753 | 0.4187 |
| | $\delta^{15}N$ | Muscle-blood | -0.1892 | 0.3025 | 0.9991 | 0.0150 | 0.0267 | 0.1634 | 0.0843 | 0.2904 |
| | | Muscle-fin | 0.2561 | 0.4765 | 0.9694 | 0.0234 | 0.0813 | 0.2851 | 0.1104 | 0.3322 |

**Table 5. Parameters and coefficients of isotope offsets linear model with the random effect of site.**

| Model parameters | | | Fixed effect | | | | Random effect | | | |
|---|---|---|---|---|---|---|---|---|---|---|
| Species | Isotope | Comparison | Intercept value | Intercept std. error | Slope value | Slope std. error | Intercept variance | Intercept std. dev. | Residual variance | residual std. dev. |
| European catfish | offset $\delta^{13}C$ | Blood- body mass | 0.1676 | 0.3225 | 0.0425 | 0.0215 | 0.2431 | 0.4931 | 0.7081 | 0.8415 |
| | | Fin- body mass | 0.7873 | 0.3025 | 0.0466 | 0.0221 | 0.1832 | 0.4280 | 0.7547 | 0.8687 |
| | offset $\delta^{15}N$ | Blood- body mass | 0.8052 | 0.1969 | -0.0137 | 0.0102 | 0.1162 | 0.3409 | 0.1572 | 0.3965 |
| | | Fin- body mass | 0.6164 | 0.1640 | -0.0022 | 0.0125 | 0.0492 | 0.2218 | 0.2421 | 0.4920 |
| Northern pike | offset $\delta^{13}C$ | Blood- body mass | -0.0351 | 0.0954 | -0.0002 | 0.0127 | 0.0223 | 0.1494 | 0.0752 | 0.2743 |
| | | Fin- body mass | 0.6724 | 0.1846 | 0.0004 | 0.0200 | 0.1016 | 0.3188 | 0.1842 | 0.4292 |
| | offset $\delta^{15}N$ | Blood- body mass | 0.2556 | 0.0945 | -0.0129 | 0.0133 | 0.0202 | 0.1421 | 0.0837 | 0.2893 |
| | | Fin- body mass | 0.3494 | 0.1353 | -0.0029 | 0.0158 | 0.0516 | 0.2271 | 0.1154 | 0.3398 |

their diet consisted almost exclusively of fish [12, 63]. Particularly, semi-aquatic prey significantly differed in the SIA signal from other food sources [12, 59]. This distinct diet composition among study sites, along with varying growth rates, which likely affect the isotopic turnover rate, could be the main reason for the offset differences between tissues at different locations. This is supported by the observation that differences were evident in nine cases between oligotrophic and meso-eutrophic sites, but only in three cases between meso-eutrophic sites and once between the oligotrophic sites. In addition to the different protein composition of tissues [37], the dietary changes during the season are probably the main factor contributing to the offset differences between tissues of individual species within one population at one study site. This is further supported by the observation that the European catfish, which exhibits higher seasonal diet plasticity than the Northern pike [12, 13], shows a more pronounced difference in offset between tissues. The better body condition of catfish in the more productive Římov and Žlutice may also have caused the higher C:N ratio variability in muscle than in blood and fin due to a potentially higher level of lipid reserves stored in muscle tissue of some predators. The muscle tissue, being metabolically active, undergoes continuous metabolic changes such as growth, repair, and energy storage [64] that can result in a more variable range of C:N ratios compared to more stable tissues like blood and fin.

To the best of our knowledge, blood has not yet been proposed as an alternative tissue to muscle, and our study is also the first ever to compare the isotope values between blood and muscle in European catfish and Northern pike. Within the studied populations of the two predatory species, the isotopic offsets between blood and muscle were usually smaller compared to the isotopic offsets observed between fin and muscle. Particularly in the case of Northern pike, isotope values between blood and muscle did not significantly differ in half of the studied populations. Even in those populations where these values differed significantly, the offset between tissues was not substantial (ranging from 0.10 to 0.43‰; Table 2). These results indicate that in certain species and populations the isotopic offsets between blood and muscle of $\delta^{13}C$ and $\delta^{15}N$ may be negligible. However, predicting routinely in which populations such a situation will occur is challenging. Thus, it is essential to asses initially the isotopic offsets between tissues for each population of interest. If necessary, a population-specific inter-tissue conversion can then be calculated.

For both species, the offsets between tissues differed significantly across some of the study sites. We calculated the conversion equations for each study site individually, both for $\delta^{13}C$ and $\delta^{15}N$ in both blood and fin of both studied species. The conversion equations from blood

to muscle and from fin to muscle were all significant. Their coefficients of determination were notably high: $R^2$ for $\delta^{13}C$ exceeded 0.77 and 0.46 for blood and fin, respectively, and for $\delta^{15}N$ it exceeded 0.87 and 0.74 for blood and fin, respectively. The isotopic relationships between tissues in individual populations of European catfish and Northern pike exhibited better results on average than those observed in other fish species [17, 22, 39, 33, 58]. This could possibly be attributed to our study's focus on adult individuals of two large fish species, as isotopic turnover rates generally decrease with body mass [30]. Thus, the $\delta^{13}C$ and $\delta^{15}N$ values between tissues may have been in better agreement in these larger adult fish. As such, the isotopic offsets between blood and muscle, as well as fin and muscle, can be used to formulate a tissue conversion model, allowing collected blood and fin tissue values to be converted to equivalent muscle tissue values for integration into food web analyses. However, establishing a species- or population-specific conversion model might require a muscle biopsy from a small number of fish, as the models for both species and both tissues varied to some extent among the study sites. A number of prior studies focusing on inter-tissue isotopic differences in other fish species also suggest that fractionation between tissues could differ among populations of the same species, potentially necessitating population-specific conversion equations [22, 39, 61, 62]. We did not observe any evident trend in the effect of body mass on the isotopic differences between tissues for either European catfish or Northern pike at any study sites. This lack of trend might indicate that inter-tissue isotope fractionations remain relatively consistent during the adult life-history stage. Regarding lipid content (approximated by C:N ratio), both species exhibited a clear pattern. The C:N ratio was consistently lowest in muscle, intermediate in fin, and highest in blood across all cases. Although significant differences in C:N ratio were evident in muscle and fin across study sites, the blood consistently displayed an unexpectedly stable C:N ratio, with no noticeable variation among study sites.

In conclusion, both $\delta^{13}C$ and $\delta^{15}N$ values for blood and fin serve as highly accurate predictors of respective isotope values for muscle tissue in adult apex predators like European catfish and Northern pike. Generally, blood proves to be a more suitable tissue than fin as it corresponds more closely to muscle tissue. Although blood sampling is time-consuming and requires technical proficiency, with practice, it is feasible to routinely draw blood from larger fish using minimally invasive methods [54]. However, the routine collection of fin clips stands out as a simpler and more commonly employed approach [17, 53, 55]. The variations in offsets among populations were notably different, not only for the studied European catfish and Northern pike but also for other species examined previously [22, 61, 62]. These differences likely stem from site-specific environmental conditions, population-related physiological factors, and mainly due to the varying diet composition. Thus, it is advisable to determine isotopic offsets between tissues from a small sample e.g. 10 individuals within each population of interest, in order to ascertain the necessary conversion correction.

Finally, our optimal guide for SIA, not only for valuable fish but ideally for any other species, suggests avoiding the need to kill individuals. Instead, if their size allows, it is preferable to conduct a muscle tissue biopsy in a necessary subsample of approximately 10 individuals from each studied locality. Concurrently, for these and all other individuals, opt for less harmful tissue sampling methods, such as blood or fin. Based on the 10 individuals, perform a regression analysis between muscle tissue and the less harmful tissues. Subsequently, these conversion-corrected data could be used. This process eliminates the need to harm or kill more individuals.

## Supporting information

**S1 File. Dataset.** The dataset analyzed during the present study.
(XLSX)

## Acknowledgments

We thank Dr. Mikko Kiljunen for help with SIA at the University of Jyväskylä, and Dr. David Hochman, a professional English editor, for English editing. We also thank the two anonymous reviewers and the editor for helpful comments and suggestions.

## Author Contributions

**Conceptualization:** Lukáš Vejřík, Ivana Vejříková.

**Funding acquisition:** Jiří Peterka, Martin Čech.

**Investigation:** Lukáš Vejřík, Zuzana Sajdlová, Luboš Kočvara, Tomáš Kolařík, Daniel Bartoň, Tomáš Jůza, Petr Blabolil, Jiří Peterka, Martin Čech, Mojmír Vašek.

**Methodology:** Lukáš Vejřík.

**Project administration:** Lukáš Vejřík.

**Visualization:** Lukáš Vejřík, Zuzana Sajdlová, Petr Blabolil.

**Writing – original draft:** Lukáš Vejřík, Ivana Vejříková.

**Writing – review & editing:** Lukáš Vejřík, Ivana Vejříková, Zuzana Sajdlová, Luboš Kočvara, Tomáš Kolařík, Daniel Bartoň, Tomáš Jůza, Petr Blabolil, Jiří Peterka, Martin Čech, Mojmír Vašek.

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
