## [Decision Letter · Decision Letter 0]

31 Oct 2023

PONE-D-23-27713A non-lethal stable isotope analysis of valued freshwater predatory fish using blood and fin tissues as alternatives to muscle tissuePLOS ONE

Dear Dr. Vejříková,

Thank you for submitting your manuscript to PLOS ONE. After careful consideration, we feel that it has merit but does not fully meet PLOS ONE’s publication criteria as it currently stands. Therefore, we invite you to submit a revised version of the manuscript that addresses the points raised during the review process.

The two reviewers agreed that your paper should submitted, pending minor revisions.  Please review the comments of the reviewers and address my requests listed below.  Before the ms can be published, you will need to address ten points raised by the reviewers:

Address these two language issues, raised by one reviewer:

       - are all three sampling methods "non lethal"?   is there high handling mortality of the fish?  

       - are the blood and muscle SI values “iinterchangeable", when they are statistically different?

Introduction:  Please add some wording (and references) relating to the seasonality of the SI values in these species and their biokinetics. Namely, a reviewer asked for information on how long it takes to change the SI value of blood, muscle, and fins. Introduction:  It would also be useful to provide some information about whether functional groups of fish (omnivorous, herbivorous, planktivorous) differ from predatory fish among tissue fractionation and biokinetics.  Additionally, are there differences between juveniles (growing rapidly) and adults (not growing rapidly)?

Analysis:  Improve the regression analysis, as suggested by one of the reviewers.

Results:  Please provide any supporting information on the food sources that could help explain the  differences between the same tissues between water bodies.  Also, please discuss these water body differences and provide an explanation for why the offsets within tissue differ among water bodies (in tables 2 and 3)

Results:  In Figure 2, the Catfish C:N ratio is rather stable in blood and fin but highly variable in muscle. Can you explain this pattern and comment on its importance for TDF, as recently suggested in the literature?

Table 2: Please clarify the caption, to ensure its clear which tissue differences are being shown.  Are these differences always between muscle tissue and a given tissue?  Additionally, please clearly spell out what are the different tissues.

Figure 2: Do not use a line connecting the box plots from different localities. Its not needed. Discussion: Please consider the reviewer’s comments regarding this statement: “Additionally, this phenomenon could be linked to the relatively similar isotopic turnover rates of muscle and blood [48].“  Some publications suggest both similarity and dissimilarity, and this is a key point of papers looking at multiple tissues. Please expand this section of the discussion to provide a broader description of these complex patterns: with similarities and dissimilarities.

Finally, one reviewer asked for a summary statement about the use of SI values from blood and fin to predict the muscle SI values.  Please elaborate this section of the conclusion and discuss what next steps would be required to validate the use of these proxies, given the results of your paper.   

We look forward to receiving your revised manuscript.

Kind regards,

David Hyrenbach, Ph.D.

Academic Editor

PLOS ONE

Journal Requirements:

3. To comply with PLOS ONE submissions requirements, in your Methods section, please provide additional information regarding the experiments involving animals and ensure you have included details on (1) methods of sacrifice, (2) methods of anesthesia and/or analgesia, and (3) efforts to alleviate suffering.

4. We note that [Figure 1] in your submission contain [map/satellite] images which may be copyrighted. All PLOS content is published under the Creative Commons Attribution License (CC BY 4.0), which means that the manuscript, images, and Supporting Information files will be freely available online, and any third party is permitted to access, download, copy, distribute, and use these materials in any way, even commercially, with proper attribution. For these reasons, we cannot publish previously copyrighted maps or satellite images created using proprietary data, such as Google software (Google Maps, Street View, and Earth). For more information, see our copyright guidelines: http://journals.plos.org/plosone/s/licenses-and-copyright.

Reviewers' comments:

Reviewer's Responses to Questions

**Comments to the Author**

1. Is the manuscript technically sound, and do the data support the conclusions?

Reviewer #1: Partly

Reviewer #2: Yes

2. Has the statistical analysis been performed appropriately and rigorously? 

Reviewer #1: Yes

Reviewer #2: Yes

3. Have the authors made all data underlying the findings in their manuscript fully available?

Reviewer #1: Yes

Reviewer #2: Yes

4. Is the manuscript presented in an intelligible fashion and written in standard English?

Reviewer #1: Yes

Reviewer #2: Yes

5. Review Comments to the Author

Reviewer #1: This manuscript is well-written, with useful findings about the use of different tissue types for SIA in two fish species and how they compare with each other. I have attached some specific comments that should be addressed about some of the language (mainly the claims of "non lethal" excluding non-lethal muscle biopsies, and "interchangeable" blood and muscle SIA when statistically different) and recommendations for improved regression analysis, which should be relatively easy to revise.

Reviewer #2: Vejříková et al. Highlight a very interesting topic in stable isotope science. Gaining the same results without killing consumers or prey sounds promising and a good idea. The manuscript is well-written without any significant problems. I have only a few comments related to biokinetic and trophic discrimination factors. Thus, I suggest a minor revision.

Specific comments

Introduction

The introduction is very well written. Nevertheless, I am missing information about how the SI signal is changed across the season in the mentioned tissues. Also, additional information on biokinetics is missing.

It is known how long it takes to change the SI value of blood, muscle, and fins. This is very important information for foodweb studies.

Also, is it known whether other functional groups of fish (omnivorous, herbivorous, planktonivorous fish) differ from predatory fish among tissue, biokinetics etc? What about the differences between juveniles and Adults? I believe that such such information should be added to the introduction.

Results

Do you have food source data to explain differences among the same tissues between water bodies? Or do you expect that there will be some additive environmental drivers that might explain the differences among ecosystems? Also, is there any explanation for why the offset in Tables 2 and 3 within tissue differs among water bodies?

Similarly, in Figure 2. In Catfish, the C:N ratio is rather stable in blood and finn but highly variable in muscle. Why is that so? It might be important for TDF because recently, it was suggested by Brauns et al. 2018 that the C:N:P ratio might be a good predictor for TDF.

TABLE 2. It is slightly difficult for me to understand what the significance belongs to. However, I assume the differences are always between muscle tissue and given tissue. If so, please add such information to the table description to be easy to follow. Also, the abbreviation of tissue should be mentioned in the table description. In addition, offset means how given tissue is enriched or depleted compared to muscle tissue, right?

In Figure 2. I would avoid to use the line connecting the box plots from different localities

Discussion

The statement

“Additionally, this phenomenon could be linked to the relatively similar isotopic turnover rates of muscle and blood [48].“

I would disagree with the statement. It is too simplified and needs to be re-made. Some publications suggest both similarity and dissimilarity. This point is crucial for the whole application of the suggested approach. Therefore, it will be good to mention it properly in the introduction and discussion.

For example, these two publications suggest dissimilarity between muscle and blood.

Buchheister and Latour 2010. Turnover and fractionation of carbon and nitrogen stable isotopes in tissues of a migratory coastal predator, summer flounder (Paralichthys dentatus)

Ankjærø 2012. Tissue-specific turnover rates and trophic enrichment of stable N and C isotopes in juvenile Atlantic cod Gadus morhua fed three different diets

However, some publications suggest that blood cells and muscles might have similar turnover rates. There is one example.

Heady and Moore 2013. Tissue turnover and stable isotope clocks to quantify resource shifts in anadromous rainbow trout

Statement in the last paragraph

“In conclusion, both δ13C and δ15N in both blood and fin serve as highly accurate predictors of those in muscle tissue for adult apex predators like European Catfish and Northern pike.“

In case you also have food source data, it will be very nice to produce MixSiar models based on data from different tissues to see the differences.

Last think related to your conclusion. I guess me and other readers will appreciate some straightforward message. Do this step, followed by another step, and you will obtain the same results as you kill the fish.

6. PLOS authors have the option to publish the peer review history of their article (what does this mean?). If published, this will include your full peer review and any attached files.

Reviewer #1: No

Reviewer #2: No

---

## [Author Response · Author response to Decision Letter 0]

4 Dec 2023

The text below can be also found in the file "PONE-D-23-27713R1_Response to Reviewers"

A point-by-point response to PONE-D-23-27713

A non-lethal stable isotope analysis of valued freshwater predatory fish using blood and fin tissues as alternatives to muscle tissue

We would like to express our gratitude to the two anonymous reviewers for their constructive comments and to the editor for providing a helpful summary of the requests. We have revised the manuscript in accordance with the feedback received. Below, you will find a point-by-point response addressing the concerns raised. All suggestions have been accepted.

Editor:

The two reviewers agreed that your paper should submitted, pending minor revisions. Please review the comments of the reviewers and address my requests listed below. Before the ms can be published, you will need to address ten points raised by the reviewers:

Address these two language issues, raised by one reviewer:

 - are all three sampling methods "non lethal"? is there high handling mortality of the fish? 

 - are the blood and muscle SI values “iinterchangeable", when they are statistically different?

We have incorporated a dedicated paragraph highlighting the advantages of blood sampling over biopsy in our manuscript. Please refer to page 4, L 67−82 for detailed information. We appreciate your valuable input, which has enhanced the clarity and citation potential of the text. 

We have rephrased a sentence in the Abstract that was misleading regarding the accurate alternative to muscle. The subsequent sections of the manuscript elaborate on the isotopic offset. 

2. Introduction: Please add some wording (and references) relating to the seasonality of the SI values in these species and their biokinetics. Namely, a reviewer asked for information on how long it takes to change the SI value of blood, muscle, and fins.

In the introduction, we have included detailed information about the turnover rate of various body tissues, considering the individual's size and temperature. Subsequently, in the discussion, we provided detailed data on the turnover rates of blood, muscle, and fins specifically calculated for the species under study in the study sites, referencing Vejřík et al. (2023). For the European catfish, the mean isotopic half-life values were 39, 44, and 153 days for blood, fin, and muscle, respectively. In the case of the Northern pike, the corresponding values were 39, 41, and 139 days for blood, fin, and muscle, respectively.

3. Introduction: It would also be useful to provide some information about whether functional groups of fish (omnivorous, herbivorous, planktivorous) differ from predatory fish among tissue fractionation and biokinetics. Additionally, are there differences between juveniles (growing rapidly) and adults (not growing rapidly)?

We have included an extensive section on the biokinetics in the introduction. We highlighted the varying turnover rates for 13C and 15N, emphasizing the significant impact of temperature (seasonal) and organism size (differences between juveniles and adults) on these rates. Lastly, we outlined the approximate order of individual tissues in terms of the mean isotopic half-life. Given our study's specific focus on two apex predators, we opted not to delve into the biokinetics related to different trophic levels. Nonetheless, we firmly believe that this topic is now extensively discussed.

1. Analysis: Improve the regression analysis, as suggested by one of the reviewers.

The analysis was improved as suggested.

5. Results: Please provide any supporting information on the food sources that could help explain the differences between the same tissues between water bodies. Also, please discuss these water body differences and provide an explanation for why the offsets within tissue differ among water bodies (in tables 2 and 3)

Unfortunately, we do not have SIA for all trophic levels of our study sites in 2017 (when we collected the data presented here). While we possess this information from other years, the SIA values of individual trophic levels can exhibit fluctuations within years. Thus, we consider it inappropriate to amalgamate them with the 2017 data. Nevertheless, we have previously published both SIA and gut content data from prior years for the apex predators and the study sites. Additionally, a publication in preparation will elucidate the food sources of these predators based on gut content analyses spanning multiple seasons. Thus, we opted not to include information about diet in the results. Instead, we intend to discuss this topic extensively in the discussion section, drawing from both our previously published studies and the upcoming study. We are confident that this additional information will significantly enhance the overall study's quality and help address various uncertainties.

6. Results: In Figure 2, the Catfish C:N ratio is rather stable in blood and fin but highly variable in muscle. Can you explain this pattern and comment on its importance for TDF, as recently suggested in the literature?

It can be observed in Figure 2 that the C/N ratio varies only in catfish from Žlutice and Římov; this ratio remains stable in catfish from Most and Milada.

We have added the discussion focused on the muscle/blood differences: “The better body condition of catfish in the more productive Římov and Žlutice reservoirs may also have caused the higher C:N ratio variability in muscle than in blood and fin due to a potentially higher level of lipid reserves stored in the muscle tissue of some predators. The muscle tissue, being metabolically active, undergoes continuous metabolic changes such as growth, repair, and energy storage [Lv et al. 2012] that can result in a more variable range of C:N ratios compared to more stable tissues like blood and fin”.

7. Table 2: Please clarify the caption, to ensure its clear which tissue differences are being shown. Are these differences always between muscle tissue and a given tissue? Additionally, please clearly spell out what are the different tissues.

While the original description of Table 2 was unclear, we have revised it, and we are confident that it is now much clearer.

8. Figure 2: Do not use a line connecting the box plots from different localities. Its not needed.

We removed the lines connecting the box plots from different localities in the figure 2.

9. Discussion: Please consider the reviewer’s comments regarding this statement: “Additionally, this phenomenon could be linked to the relatively similar isotopic turnover rates of muscle and blood [48].“ Some publications suggest both similarity and dissimilarity, and this is a key point of papers looking at multiple tissues. Please expand this section of the discussion to provide a broader description of these complex patterns: with similarities and dissimilarities.

The reviewer's comments are relevant, emphasizing the crucial role of the biokinetics of stable isotopes in determining the final SI signal of individual tissues. It is indeed true that scientific conclusions in this regard are not entirely unambiguous. Consequently, we have augmented the Introduction with a comprehensive part on the biokinetics of stable isotopes in individual tissues. Moreover, we have made significant changes to and clarified statements in the discussion section.

10. Finally, one reviewer asked for a summary statement about the use of SI values from blood and fin to predict the muscle SI values. Please elaborate this section of the conclusion and discuss what next steps would be required to validate the use of these proxies, given the results of your paper. 

In conclusion, we have clearly expressed our opinion and provided recommendations on the ideal approach for analyzing stable isotopes in fish. This ensures that such analyses are no longer necessarily linked to the sacrifice of individuals, simultaneously minimizing the degree of harm inflicted."

Additional requirements

1. PLOS ONE's style requirements

The style requirements were updated according to the templates.

2. Permits obtained for the work

The permission is mentioned, Lines 129-131.

3. Additional information regarding the experiments

The anesthesia was specified: a bath containing clove oil. See L 153.

4. Figure copyright

The Figure 1 was made by Zuzana Sajdlova (the coauthor) in the Inkscape freeware program specifically for this manuscript, thus no copyright is needed.

5. References

The references were updated.

 

PLOS One Vejrik et al 2023 Review

“A non-lethal stable isotope analysis of valued freshwater predatory fish using blood and fin tissues as alternatives to muscle tissue”

Reviewer #1: This manuscript is well-written, with useful findings about the use of different tissue types for SIA in two fish species and how they compare with each other. I have attached some specific comments that should be addressed about some of the language (mainly the claims of "non lethal" excluding non-lethal muscle biopsies, and "interchangeable" blood and muscle SIA when statistically different) and recommendations for improved regression analysis, which should be relatively easy to revise.

Abstract:

- Can you say that the blood and fin clips are less invasive to the muscle biopsy if none of the approaches were lethal to the fish? Are there studies you can cite in the introduction or discussion that examined fish response and performance after each sampling technique was used to support this claim of “less invasive”? 

We have incorporated a dedicated paragraph highlighting the advantages of blood sampling over biopsy in our manuscript. Please refer to page 4, L 67−82 for detailed information. We added the appropriate references.

- What do you mean by “blood proved to be more accurate alternative”? In what sense is it more accurate – does it correspond better to the muscle? Change “more accurate” to “provided better correspondence to muscle isotope values”

We rephrased the part according to the suggestion, Thank you.

- How can blood and muscle “be used interchangeably” but they also “vary significantly” as seen in the ANOVAs? 

The incorrect wording was removed. 

Introduction: 

- Remove period after “Approximately” in first paragraph.

- 

Removed

- More elaboration on why blood sampling is preferable to non-lethal muscle biopsy would be useful in paragraph 2. 

- 

Thank you, we have incorporated a paragraph highlighting the advantages of blood sampling over biopsy, see L 67−82.

- Reorder citations in paragraph 2 [31,17] �[17, 31]

Corrected

Methods:

- “samples of tree body tissues were collected” should be “three”

Corrected

- Include your alpha value in data analysis

The alpha value (standard level of significance of α = 0.05) was added in „Data analysis“ description.

- Need to include what diagnostics you performed for the linear regression analysis & please state whether you examined your residuals 

We have improved the description of linear regression analysis as suggested.

Results:

- Remove “statistically” from all “statistically significant”, it is redundant

Removed

- Report SE instead of SD in table 2

We replaced SD with SE

- Would not say “The blood generally showed only slight enrichment in D13C compared to the muscle tissue” given that it was 50/50 enrichment/depletion in table 2. 

- 

"Yes, that's correct. We have rephrased the sentence to better reflect the situation. 

- Edit: “The degree of these differences was more pronounced in the blood compared with fins of European catfish” In this sentence what do you mean by “it was lower for Northern Pike”? The degree of differences between blood and fin look similar in the two species.

Edited. We thought that the degree of differences between blood and fin was less noticeable in pike. We have reformulated the sentence.

- Last sentence of paragraph 2 calls the blood and fin samples “non-lethal” tissue, but the muscle sampling did not kill the fish either and would therefore also be considered “non-lethal”. Throughout the text, the blood and fish samples are called “non lethal” which is an unfair distinction from the biopsy that was also non lethal. 

Yes, that is accurate. For this reason, we designated fin clips and blood as less harmful rather than non-lethal. In the introduction, we explained the distinctions in the sampling methods for these various tissues. 

- First sentence in paragraph 3: remove “the” in “the both D13C and D15N offsets”

Removed

- Paragraph after table 3: “however the difference between fin and muscle was statistically insignificant” � “however no difference was found between fin and muscle” 

Thank you, replaced

- Change any phrases of “difference was insignificant” to “no difference was found” or “there was no difference”

Replaced

- Consider using a linear model with the random effect of site instead of separate regressions for each site (if you are interested in calculating an overall conversion equation for each fish and tissue type). If you want a conversion equation for each lake then fixed effect of site. 

As suggested, the linear model with the random effect of site was applied to model the relationships between mussel isotopes with the isotopes in fin and blood and between the isotope offsets and fish mass. The models were prepared for each species and isotope C and N separately. After fitting the models, regression diagnostics was applied by plotting the residual vs. fitted values and Normal QQ Plot to check the residuals variance.

- Last paragraph of results rephrase first two sentences: “No distinct trends were found through linear regression analysis between isotopic tissue offsets and fish body mass in either species or in any study site.”

Replaced, thank you

- I recommend again running a random effect linear model on the tissue offsets and body mass (random effect of site) and reporting the test statistics from those model runs rather than averaging R squared values from multiple different linear regressions.

- Need to report the results of model regression diagnostics. 

The linear model was improved as suggested and the Table with results was added. 

Discussion:

- From my understanding it seems like the muscle biopsy is not lethal to moderate-large sized predatory fish, and so the blood/fin vs muscle argument may be redundant. How does blood sampling affect smaller fish, where taking a muscle biopsy can be particularly invasive/lethal? Does blood sampling impose any negative effects on the fish compared to a biopsy punch? What is the advantage of blood sampling when the biopsy punch is already non-lethal approach to white muscle samples?

We have emphasized the benefits of blood sampling over biopsy. For detailed information on sample size, healing time, and the safe body mass for blood drawing in fish, please see page 4, L 67−82. The method is deemed safe for fish with a body mass as small as 200 g.

- Can you include general turnover rates of each of the tissue types sampled? 

In the discussion, we incorporated the isotopic half-life values under actual conditions for both an average-sized catfish and an average-sized pike across all four studied tissues. The full turnover is then approximately four to five times longer.

- Any inferences on why your fin D15N values were enriched compared to muscle? 

We supplemented the discussion with examples where, similar to ours, it was observed that fin d15N values were enriched compared to muscle. Thus, it seems that this phenomenon is species-specific and far from being as clear as in the case of d13C, although according to the studies so far, the prevailing observation is that fin is depleted in d15N compared to muscle.

- Bold claim to make that “blood and muscle tissues can be used interchangeably in SIA without requiring conversion”, given that you showed that the isotope offsets were significant in 5/8 of your paired t-tests for d13C and 6/8 for d15N, and linear regression models were all significant. This claim is not substantiated.

Yes you are right. We have reformulated this statement.

- “Thus the D13C and D15N values between tissues may have been in better agreement in these larger adult fish” instead of “were apparently more stable over time”

Replaced

- “In conclusion, both d13C and d15N values for blood and fin serve as highly accurate predictors of respective isotope values for muscle tissue in adult apex predators…”

Thank you, replaced

- Change “and exhibits a closer resemblance” to “as it corresponds more closely” 

Replaced

- Remove “significant” in “were notably significant different” 

Removed

Thank you very much for the valuable advice, which we have all incorporated into the manuscript. 

 

Reviewer #2: Vejříková et al. Highlight a very interesting topic in stable isotope science. Gaining the same results without killing consumers or prey sounds promising and a good idea. The manuscript is well-written without any significant problems. I have only a few comments related to biokinetic and trophic discrimination factors. Thus, I suggest a minor revision.

Specific comments

Introduction

The introduction is very well written. Nevertheless, I am missing information about how the SI signal is changed across the season in the mentioned tissues. Also, additional information on biokinetics is missing. It is known how long it takes to change the SI value of blood, muscle, and fins. This is very important information for foodweb studies. Also, is it known whether other functional groups of fish (omnivorous, herbivorous, planktonivorous fish) differ from predatory fish among tissue, biokinetics etc? What about the differences between juveniles and Adults? I believe that such such information should be added to the introduction.

Thank you for bringing attention to this important issue. We have added an extensive section on this matter to the introduction, addressing the different isotopic half-life for 13C and 15N. Additionally, we discussed the significant influence of ambient temperature and organism size on the rates. Last but not least, we outlined the approximate order of individual body tissues in terms of mean isotopic half-life. Given our study's focus on two apex predators, we made a deliberate choice not to delve into the biokinetics related to different trophic levels.

Results

Do you have food source data to explain differences among the same tissues between water bodies? Or do you expect that there will be some additive environmental drivers that might explain the differences among ecosystems? Also, is there any explanation for why the offset in Tables 2 and 3 within tissue differs among water bodies?

Thank you for your helpful note. We thoroughly examined the diet of the presented species (European catfish and Northern pike) in the specified locations, delving into the details through both Stable Isotope Analysis (SIA) and stomach content analysis within the season in two of our studies (Vejřík et al., 2017, and Říha in prep). It is indeed true that the food of these species varies significantly within localities and throughout the season. This variability is likely the primary factor contributing to the differences in offset between tissues and among localities. Consequently, we have included a detailed research analysis of this issue in the Discussion. 

Similarly, in Figure 2. In Catfish, the C:N ratio is rather stable in blood and finn but highly variable in muscle. Why is that so? It might be important for TDF because recently, it was suggested by Brauns et al. 2018 that the C:N:P ratio might be a good predictor for TDF.

We have added the discussion focused on the muscle/blood differences: “The better body condition of catfish in the more productive Římov and Žlutice reservoirs may also have caused the higher C:N ratio variability in muscle than in blood and fin due to a potentially higher level of lipid reserves stored in the muscle tissue of some predators. The muscle tissue, being metabolically active, undergoes continuous metabolic changes such as growth, repair, and energy storage [Lv et al. 2012] that can result in a more variable range of C:N ratios compared to more stable tissues like blood and fin.

TABLE 2. It is slightly difficult for me to understand what the significance belongs to. However, I assume the differences are always between muscle tissue and given tissue. If so, please add such information to the table description to be easy to follow. Also, the abbreviation of tissue should be mentioned in the table description. In addition, offset means how given tissue is enriched or depleted compared to muscle tissue, right?

We have added more detailed information about the offset to the table description. The table description has been completely reformulated, and we firmly believe that it is now clearer.

In Figure 2. I would avoid to use the line connecting the box plots from different localities

We removed the lines connecting the box plots from different localities in the figure 2.

Discussion

The statement

“Additionally, this phenomenon could be linked to the relatively similar isotopic turnover rates of muscle and blood [48].“ I would disagree with the statement. It is too simplified and needs to be re-made. Some publications suggest both similarity and dissimilarity. This point is crucial for the whole application of the suggested approach. Therefore, it will be good to mention it properly in the introduction and discussion.

For example, these two publications suggest dissimilarity between muscle and blood.

Buchheister and Latour 2010. Turnover and fractionation of carbon and nitrogen stable isotopes in tissues of a migratory coastal predator, summer flounder (Paralichthys dentatus)

Ankjærø 2012. Tissue-specific turnover rates and trophic enrichment of stable N and C isotopes in juvenile Atlantic cod Gadus morhua fed three different diets

 However, some publications suggest that blood cells and muscles might have similar turnover rates. There is one example.

Heady and Moore 2013. Tissue turnover and stable isotope clocks to quantify resource shifts in anadromous rainbow trout

Statement in the last paragraph

Yes, it is true that the issue of biokinetics is rather complex and intricate. Hence, we extensively developed this topic, especially in the Introduction. In the discussion, we delved into the biokinetics of stable isotopes within the studied species and locations, employing relatively sophisticated calculations. We highlighted a similar half-life for blood and fins, while observing a significantly longer half-life for muscle in both Northern pike and European catfish.

“In conclusion, both δ13C and δ15N in both blood and fin serve as highly accurate predictors of those in muscle tissue for adult apex predators like European Catfish and Northern pike.“

In case you also have food source data, it will be very nice to produce MixSiar models based on data from different tissues to see the differences.

Unfortunately, we do not have SIA food source data from the year 2017 available for the study sites, and it is not entirely appropriate to use data from earlier years, as they may change over time. However, we believe that we have addressed this limitation by conducting a relatively detailed discussion based on our two studies, focusing on the diet of Northern pike and European catfish in the study sites. The study by Vejřík et al. 2017 even directly contains the proposed MixSiar models for both species."

Last think related to your conclusion. I guess me and other readers will appreciate some straightforward message. Do this step, followed by another step, and you will obtain the same results as you kill the fish.

Thanks for the very helpful note. In conclusion, we have added our clear opinion and recommendations on how to ideally proceed SIA in fish. This ensures that such analyses are no longer necessarily linked to the sacrifice of individuals, simultaneously minimizing the degree of harm inflicted.

---

## [Editor Report · Decision Letter 1]

27 Dec 2023

A non-lethal stable isotope analysis of valued freshwater predatory fish using blood and fin tissues as alternatives to muscle tissue

PONE-D-23-27713R1

Dear Dr. Vejříková,

We’re pleased to inform you that your manuscript has been judged scientifically suitable for publication and will be formally accepted for publication once it meets all outstanding technical requirements.  We  thank you for addressing the reviewer comments completely and thoroughly.   

Kind regards,

David Hyrenbach, Ph.D.

Academic Editor

PLOS ONE

---

## [Editor Report · Acceptance letter]

9 Jan 2024

PONE-D-23-27713R1 

PLOS ONE

Dear Dr. Vejříková, 

I'm pleased to inform you that your manuscript has been deemed suitable for publication in PLOS ONE. Congratulations! Your manuscript is now being handed over to our production team.

Kind regards, 

on behalf of

Dr. David Hyrenbach 

Academic Editor

PLOS ONE